# Differences in family planning outcomes between military and general populations in Kinshasa, Democratic Republic of the Congo: a cross-sectional analysis

Pierre Akilimali,[1] Philip Anglewicz,[2] Henri Nzuka Engale,[3]
Gilbert Kabanda Kurhenga,[3] Julie Hernandez,[2] Patrick Kayembe,[1] Jane Bertrand[2]

¹Kinshasa School of Public Health, Universite de Kinshasa, Kinshasa, The Democratic Republic of the Congo
²School of Public Health and Tropical Medicine, Tulane University, New Orleans, Louisiana, USA
³Medical Division, Congolese Armed Forces, Kinshasa, The Democratic Republic of the Congo

**Correspondence to**
Dr Philip Anglewicz;
panglewi@tulane.edu

## ABSTRACT

**Objectives** To examine family planning outcomes among women living in military camps in Kinshasa, Democratic Republic of the Congo, and compare these outcomes with a representative sample of non-military women in Kinshasa.

**Participants** Women of reproductive ages, 15–49 years. We compare two populations: women living in military camps and the general (non-military) population in Kinshasa.

**Study design** For sampling, we used a two-stage cluster sampling design, where we first randomly selected enumeration areas (EA), and then randomly selected women within each EA (separately for each of the two populations). We administered a survey on contraceptive use and family planning to all participating women. We use bivariate and multivariate analysis to compare these populations for a range of family planning outcomes.

**Results** We find many statistically significant differences between women in military camps and general female population of Kinshasa. Although they do not have more children, women in military camps are less likely to be using contraception (all methods OR 0.24, 95% CI 0.11 to 0.53; modern methods OR 0.25, 95% CI 0.08 to 0.79; traditional methods OR 0.41, 95% CI 0.24 to 0.71) and less knowledgeable about many family planning methods (less likely to have heard of implants (OR 0.23, 95% CI 0.11 to 0.48), injectables (OR 0.19, 95% CI 0.08 to 0.44), condoms (OR 0.23, 95% CI 0.12 to 0.47), withdrawal (OR 0.05, 95% CI 0.02 to 0.17) and rhythm (OR 0.12, 95% CI 0.03 to 0.44) methods), while at the same time they are more likely to want to limit their births (OR 5.17, 95% CI 2.52 to 10.62), and less likely to have obtained their preferred family planning method (OR 0.14, 95% CI 0.03 to 0.64).

**Conclusions** Women in military camps in Kinshasa appear to be an important and underserved population with regard to family planning. Our results suggest that women in military camps have limited access to modern family planning methods.

## INTRODUCTION

Research on family planning practices among the military population in low-income and middle-income countries is limited. During

### Strengths and limitations of this study

► Examine family planning outcomes for an important but understudied population, women in military camps in a high-fertility setting.
► Use representative data for both military camp and general population in Kinshasa.
► Lacking some measures importantly related to family planning in this environment.

the HIV/AIDS epidemic, the military was seen as a population at particular risk of infection and was therefore the topic of HIV-related research.[1–8] But research on family planning among the military in developing settings has been conducted in few geographic areas, such as India[9] and Nigeria.[8 10–13] Furthermore, the scant existing research has important limitations, such as the lack of a comparison group from the general population, so it is often not known if the military population is at particular need of family planning programmes.[8–13]

In order to increase knowledge and use of contraceptive methods, programmes often target specific subpopulations that have lower contraceptive use and access, such as poor, urban or adolescent female populations.[14–16] These populations then become the subjects for targeted family planning programmes, in order to improve overall population-level fertility and family planning outcomes. However, some groups that may differ in family planning outcomes are seldom examined; the military is one such population.

There is reason to believe that the military population is highly relevant for family planning research and programmes. In some settings, the military population composes a non-negligible percentage of the overall population and may therefore have a noticeable impact on aggregate family planning measures. The military also may have

different family planning practices than the general population: research in the USA has shown that the military population has relatively higher fertility and earlier family formation.[17]

The relationship between military participation and fertility is ambiguous. Some research has shown that the military is an environment that promotes family formation, and military members marry earlier and higher fertility.[17] Benefits offered for military families can facilitate childbearing.[17] As a result, it is perhaps not surprising that most studies show high rates of pregnancy and low contraceptive use among military in the USA.[18–21]

There are also reasons why one might expect lower fertility and greater contraceptive use among female members of the military. Pregnancy inhibits the ability to train for and serve in active duty,[22] and has led to evacuation from military activity for female troops.[23] Military activity may separate spouses for extended periods, thereby limiting opportunity for childbearing.[24] Military service can also cause stress within marriage, which may impact fertility.[24] Some studies have found higher use of oral contraceptive in the military.[22]

In this research, we examine family planning-related outcomes among women in military camps in Kinshasa, Democratic Republic of the Congo (DRC). We focus on a range of family planning outcomes, such as fertility, contraceptive use (overall, modern and traditional), whether the last birth was unintended, desire for an additional child, family planning knowledge and exposure to family planning messages. We compare family planning outcomes for the female military camp population in Kinshasa with a comparable survey of that is representative of the general, non-military female population in Kinshasa.

## METHODS
### Setting
The DRC is Africa's fourth most-populous and one of the region's fastest growing countries.[25] DRC has one of the highest fertility rates in the world: the most recent Demographic and Health Survey (DHS) from 2013 to 2014 estimated a country-level TFR of 6.6, a slight increase since the 6.3 TFR estimated from the 2007 DHS.[26 27] At the same time, contraceptive use is low in DRC: the modern contraceptive prevalence rate among women aged 15–49 years who are married or in union is 7.8% for the country as a whole, 19.0% in Kinshasa and 17.2% in Kongo Central.[27]

Military camps are located throughout the city of Kinshasa. According to government documents obtained by the authors, there are 17 military camps in Kinshasa, in which both enlisted military and their families reside. According to 2016 estimates, the total population of these camps is 305 405 which represents approximately 3% of the population of Kinshasa. The vast majority of enlisted military are male; only 4% of military members are women. Military camps

are closed environments and difficult to access by the civilian population, as such they are often not targeted by mass health activities. Although the majority of enlisted military live in camps, because housing and other services are free, not all members of the military reside there; officers in the military often live outside. All military camps have a health centre, which can be used by military members and their families.

### Data
We use two sources of data for this analysis. Our first source is the Performance Monitoring and Accountability 2020 Project (PMA2020), which was established in part to measure uptake of contraceptive use in many of the world's most populous countries (http://www.pma2020. org/). To achieve this aim, PMA2020 collects representative data in 11 countries on an annual basis for a range of fertility and family planning-related measures.

PMA2020 has collected data from two provinces in DRC, six rounds of data in Kinshasa (2013–2017) and three rounds in Kongo Central (2015–2017), a province to the west of Kinshasa. The sampling framework uses a two-stage cluster sampling approach, in which the study first randomly selects census enumeration areas (EAs) within each province, then conducts a listing of all households in these EAs, and randomly selects 33 households within each EA. PMA2020 first administers a household survey to the head of household, and then all resident women of reproductive age (15–49 years) within the household are selected for interview. The PMA2020 female survey includes basic demographic information and extensive information on fertility history and preferences, and contraceptive use. In 2016, PMA2020 interviewed 2607 women in its fifth round of data collection in Kinshasa. Interviews were conducted only with female interviewers.

Data collection for the military sample also took place in 2016. In sampling this population, we used a similar two-stage cluster design approach. Out of the 17 military camps in Kinshasa, 10 were randomly drawn, proportionate to population size. These 10 military camps were then divided into EAs, and one EA was randomly drawn in each of the 10 camps. As with PMA2020, interviewers first conducted a listing of households in each EA, after which 33 households in each EA were selected for interview (with all residing women aged 15–49 years sought for interview). A total of 514 women were interviewed in the military sample. Of these women, the majority were spouses of men in the military (78.6%), while 16.6% were military members themselves. The remaining women were mothers of a military member.

All participating women provided written and informed consent to take part in this study. In this setting, individuals aged 15–17 years are considered adults, so parental consent was not necessary to interview women of these ages.

## Measures

We focus on several categories of family planning outcomes, starting with four fertility-related outcomes: the number of lifetime births, whether the woman experienced the death of one of her children, whether the women does not want another child and whether the last birth was unintended. Next, we turn to family planning use, measured as overall contraceptive use, then separated into modern and traditional methods (rhythm, withdrawal and other traditional method (folkloric methods like amulets, herbs, etc). Among those using contraception, we examine whether women obtained their desired method, phrased as "During that visit, did you obtain the method you wanted to delay or avoid getting pregnant?" Finally, we measure family planning awareness and exposure; whether the respondent had heard of injectables, implants, condoms, withdrawal and rhythm methods and whether the woman had visited a health facility in the past 12 months, heard about family planning on the radio, read about family planning in the newspaper or saw family planning billboard.

We also examine demographic characteristics for women in military camps and general female population in Kinshasa, including age, number of lifetime births, marital status, level of education and household wealth. Household wealth is measured using a constructed wealth index based on ownership of 25 household durable assets, house and roof material, livestock ownership and water source. A wealth index was created using principal component analysis,[28] which is then converted into quintiles. Survey instruments are included as online supplementary files.

## Analysis

First, we tabulate background and family planning characteristics for women in military camps and the female non-military population of Kinshasa. We account for the sampling design from data collection, using weights and accounting for clustering within EAs.

Next, we examine whether family planning differences persist, after controlling for background characteristics that may differ between women in military camps and the non-military population. We use multivariate regressions where the dependent variables are measures of interest for family planning, including number of lifetime births, use of contraception (overall contraceptives, modern contraceptives, traditional methods), not wanting another child, experiencing child mortality and last birth unintended. Independent variables include age, a quadratic term for age, level of education, marital status and household wealth quintile. To account for the study design, we cluster SEs by EA. Missing values are considered missing at random.

## Patient and public involvement

Patients or public were not involved in the study design, development of the research question, recruitment into or conduct of the study or outcome measures. Results will not be distributed to the participants themselves.

## RESULTS

Background characteristics (weighted) for women from both rounds of PMA2020 are shown in table 1, including statistical tests for differences between women in military camps and the non-military population. Generally, women in military camps and the non-military population are similar in their demographic profile, there are no statistically significant differences in background characteristics.

Although the populations are similar in demographic characteristics, they are very different in family planning-related outcomes (table 1). Women in military camps are significantly less likely to use contraception, and are significantly less likely to obtain their desired contraceptive method. As shown in table 1, women in military camps are more likely to not want another child, and to claim that their last birth was unintended. Women in military camps are less informed of family planning methods; they are significantly less likely to have heard of implants, injectables, condoms, withdrawal and rhythm. However, results for exposure to family planning programmes are mixed: they are less likely to have visited a health facility in the past 12 months but more likely to have heard about family planning on the radio, read about it in a newspaper and saw a family planning billboard.

After controlling for demographic characteristics, the multivariate results are similar to the bivariate differences between women in military camps and the non-military population. As shown in table 2, there is no statistically significant difference in number of lifetime births between women in military camps and the non-military population. However, women in military camps are significantly less likely to have experienced child mortality than the non-military (OR 0.53, 95% CI 0.33 to 0.87). Women in military camps are significantly more likely to report that they do not want another child (OR 5.17, 95% CI 2.52 to 10.62), and to report that their last birth was unintended (OR 5.19, 95% CI 2.31 to 11.62).

Results in table 3 confirm the differences in contraceptive use for these populations, in which women in military camps are significantly less likely to be using contraception of all types, modern methods and traditional methods (all methods OR 0.24, 95% CI 0.11 to 0.53; modern methods OR 0.25, 95% CI 0.08 to 0.79; traditional methods OR 0.41, 95% CI 0.24 to 0.71). Women in military camps were also significantly less likely to have obtained their desired family planning method (OR 0.14, 95% CI 0.03 to 0.64).

We also find consistent differences in knowledge of various contraceptive methods (table 4), both modern and traditional. Women in military camps are significantly less likely to have heard of implants (OR 0.23, 95% CI 0.11 to 0.48), injectables (OR 0.19, 95% CI 0.08 to 0.44), condoms (OR 0.23, 95% CI 0.12 to 0.47), withdrawal (OR 0.05, 95% CI 0.02 to 0.17) and rhythm (OR 0.12, 95% CI 0.03 to 0.44) methods.

Results for exposure to family planning are shown in table 5. Women in military camps are less likely to have visited a health facility in the past 12 months (OR 0.28,

**Table 1** Weighted background characteristics for military and non-military women, Kinshasa 2016

| | Non-military | Military |
|---|---|---|
| **Age groups (years)** | | |
| 15–20 | 21.60% | 24.48% |
| 21–25 | 20.94% | 20.02% |
| 26–30 | 16.69% | 18.42% |
| 31–35 | 14.42% | 15.85% |
| 36–40 | 11.54% | 10.51% |
| 41–45 | 8.53% | 5.87% |
| 46–49 | 6.28% | 4.85% |
| **Level of education** | | |
| No education | 2.16% | 4.31% |
| Primary | 18.55% | 13.53% |
| Middle secondary school | 63.56% | 66.76% |
| Advanced secondary or higher | 15.72% | 15.40% |
| **Marital status** | | |
| Married/partnered | 47.03% | 48.16% |
| Separated/divorced/widowed | 5.32% | 4.75% |
| Never married | 47.65% | 47.09% |
| **Wealth index** | | |
| Quintile 1 (lowest) | 15.09% | 26.43% |
| Quintile 2 | 19.33% | 27.02% |
| Quintile 3 | 21.28% | 20.65% |
| Quintile 4 | 21.33% | 15.05% |
| Quintile 5 (highest) | 22.97% | 10.85% |
| **Fertility and children** | | |
| Number lifetime births | 1.82 | 1.71 |
| Experienced child mortality | 19.65% | 12.44%* |
| Do not want another child | 18.03% | 46.72%** |
| Last birth unintended | 7.36% | 26.29%** |
| **Family planning use** | | |
| Contraceptive use | 42.28% | 17.43%** |
| Traditional contraceptive use | 21.39% | 6.56%* |
| Modern contraceptive use | 20.89% | 10.87%** |
| Obtained the desired family planning method | 92.41% | 68.25%** |
| **Family planning knowledge** | | |
| Heard of implants | 83.27% | 55.82%** |
| Heard of injectables | 87.83% | 60.91%** |
| Heard of condoms | 94.73% | 81.91%** |
| Heard of withdrawal | 80.83% | 46.94%** |
| Heard of rhythm | 91.50% | 44.57%** |
| **Family planning exposure** | | |
| Visited health facility in past 12 months | 55.18% | 28.84%** |

Continued

**Table 1** Continued

| | Non-military | Military |
|---|---|---|
| Heard about family planning on the radio | 34.65% | 58.58%* |
| Read about family planning in a magazine/newspaper | 13.12% | 50.76%** |
| Saw family planning billboard advertisement | 46.12% | 89.34%** |
| N | 2607 | 514 |

Difference between military and non-military significant at *p≤0.05, **p≤0.01.

95% CI 0.14 to 0.55). However, they are significantly more likely to have heard about family planning on the radio (OR 2.81, 95% CI 1.34 to 5.92), read about family planning in a newspaper or magazine (OR 7.72, 95% CI 3.13 to 19.04) and saw a family planning billboard (OR 9.71, 95% CI 3.63 to 25.20).

## DISCUSSION

Although the female population in military camps is similar to the general population of women in Kinshasa in background characteristics, we find many strong and highly statistically significant differences in family planning-related outcomes. Overall, these results suggest that the population residing in military camps is an important, underserved population in Kinshasa: the female population in military camps are more likely to want to limit births but are less likely to be using contraception, and less likely to have obtained their preferred family planning method. Women in military camps are more likely to be exposed to family planning messages (radio, billboard, newspaper/magazine), but are less knowledgeable about many family planning methods.

These results suggest that women in military camps have limited access to modern family planning methods. Lack of access would explain why they are less likely to use despite being more likely to want to limit births. Access limitations are also evident in the fact that women in military camps are less likely to have obtained their preferred family planning method, and are less likely to have visited a health facility in the past 12 months. Prior to this study, leadership in the DRC military (coauthors on this paper) recognised a lack of access to and limited use of modern contraception among the military population. They actively sought out a research partner to document this trend and a service delivery organisation that could work with them in improving contraceptive services within the military camps of Kinshasa and Kongo Central. Our study confirms the expectations of the DRC military that contraceptive use is lower among women in military camps, due at least in part to limited access to modern family planning.

**Table 2** Weighted regression results for differences in fertility and children between military and non-military, Kinshasa 2016

| | Number of births | | Child mortality | | Do not want another | | Last birth unintended | |
|---|---|---|---|---|---|---|---|---|
| | Coef | 95% CI | OR | 95% CI | OR | 95% CI | OR | 95% CI |
| Military | **−0.05** | **−0.31 to 0.22** | **0.53*** | **0.33 to 0.87** | **5.17**** | **2.52 to 10.62** | **5.19**** | **2.31 to 11.62** |
| Number of births | – | – | 1.16** | 1.05 to 1.29 | 1.55** | 1.31 to 1.84 | 1.43** | 1.23 to 1.66 |
| Age | 0.33** | 0.28 to 0.37 | 1.03** | 1.01 to 1.06 | 1.02* | 1.00 to 1.05 | 0.99 | 0.96 to 1.03 |
| Age squared | −0.01** | −0.00 to 0.00 | – | – | – | – | – | – |
| Level of education | | | | | | | | |
| No education (ref.) | – | – | – | – | – | – | – | – |
| Primary | 0.13 | −0.13 to 0.38 | 1.05 | 0.50 to 2.20 | 0.57 | 0.14 to 1.07 | 1.01 | 0.14 to 1.07 |
| Middle secondary school | 0.03 | −0.28 to 0.33 | 0.81 | 0.34 to 1.93 | 0.48 | 0.20 to 1.16 | 0.51 | 0.19 to 1.32 |
| Advanced secondary or higher | −0.37* | −0.73, to 0.02 | 0.43 | 0.16 to 1.18 | 0.39 | 0.14 to 1.07 | 0.67 | 0.23 to 1.98 |
| Marital status | | | | | | | | |
| Married/partnered (ref.) | – | – | – | – | – | – | – | – |
| Separated/divorced/widowed | −0.16** | −0.28 to 0.05 | 0.72 | 0.40 to 1.29 | 0.88 | 0.51 to 1.53 | 2.82** | 1.47 to 5.42 |
| Never married | −1.06** | −1.32 to 0.80 | 1.03 | 0.67 to 1.59 | 1.06 | 0.64 to 1.75 | 2.81* | 1.19 to 6.66 |
| Household wealth | | | | | | | | |
| Quintile 1 (lowest, ref.) | – | – | – | – | – | – | – | – |
| Quintile 2 | −0.13 | −0.27 to 0.01 | 0.85 | 0.48 to 1.50 | 0.94 | 0.52 to 1.63 | 0.54 | 0.20 to 1.48 |
| Quintile 3 | −0.16* | −0.31 to 0.01 | 0.88 | 0.57 to 1.37 | 0.92 | 0.19 to 1.26 | 0.69 | 0.26 to 1.83 |
| Quintile 4 | −0.12 | −0.29 to 0.04 | 0.67 | 0.41 to 1.10 | 0.48 | 0.31 to 1.22 | 1.01 | 0.47 to 2.16 |
| Quintile 5 (highest) | −0.23** | −0.40 to 0.06 | 0.77 | 0.44 to 1.35 | 0.61 | 0.50 to 1.64 | 0.82 | 0.30 to 2.27 |
| N | 3085 | | 1795 | | 2830 | | 1813 | |

The bold font is to emphasize the key independent variable in each table, which is the binary indicator for military camp population.

*P≤0.05, **p≤0.01; child mortality and last birth unintended are limited to women who have ever given birth.

**Table 3** Weighted logistic regression results for differences in family planning use for military, Kinshasa 2016

| | Using contraception | | Modern contraception | | Traditional contraception | | Obtained desired method | |
|---|---|---|---|---|---|---|---|---|
| | OR | 95% CI | OR | 95% CI | OR | 95% CI | OR | 95% CI |
| Military | **0.24**** | **0.11 to 0.53** | **0.25*** | **0.08 to 0.79** | **0.41**** | **0.24 to 0.71** | **0.14**** | **0.03 to 0.64** |
| Number of births | 1.29** | 1.17 to 1.43 | 1.16** | 1.05 to 1.28 | 1.22** | 1.13 to 1.32 | 0.98 | 0.80 to 1.20 |
| Age | 1.53** | 1.43 to 1.64 | 1.23** | 1.14 to 1.32 | 1.60** | 1.46 to 1.75 | 0.99 | 0.77 to 1.28 |
| Age quadratic | 0.99** | 0.99 to 0.99 | 0.99** | 0.99 to 0.99 | 0.99** | 0.99 to 0.99 | 1.00 | 0.99 to 1.00 |
| Level of education | | | | | | | | |
| No education (ref.) | – | – | – | – | – | – | – | – |
| Primary | 1.68 | 0.75 to 3.78 | 0.70 | 0.35 to 1.41 | 3.41* | 1.24 to 9.33 | 1.59 | 0.25 to 9.95 |
| Middle secondary school | 2.33* | 1.03 to 5.27 | 1.16 | 0.55 to 2.45 | 3.42* | 1.30 to 9.00 | 1.69 | 0.27 to 10.63 |
| Advanced secondary or higher | 3.16** | 1.33 to 7.53 | 1.55 | 0.65 to 3.71 | 3.98** | 1.44 to 11.02 | 1.13 | 0.16 to 8.00 |
| Marital status | | | | | | | | |
| Married/partnered (ref.) | – | – | – | – | – | – | – | – |
| Separated/divorced/widowed | 0.45** | 0.27 to 0.75 | 0.41* | 0.20 to 0.81 | 0.74 | 0.46 to 1.19 | 0.77 | 0.35 to 1.71 |
| Never married | 1.52* | 1.10 to 2.11 | 1.16 | 0.83 to 1.63 | 1.55** | 1.12 to 2.13 | 1.34 | 0.75 to 2.39 |
| Household wealth | | | | | | | | |
| Quintile 1 (lowest, ref.) | – | – | – | – | – | – | – | – |
| Quintile 2 | 0.91 | 0.45 to 1.83 | 1.27 | 0.70 to 2.30 | 0.73 | 0.41 to 1.30 | 0.46 | 0.13 to 1.70 |
| Quintile 3 | 0.82 | 0.41 to 1.66 | 1.17 | 0.57 to 2.38 | 0.68 | 0.41 to 1.14 | 0.45 | 0.11 to 1.77 |
| Quintile 4 | 0.99 | 0.47 to 2.11 | 1.29 | 0.60 to 2.79 | 0.81 | 0.50 to 1.33 | 0.28 | 0.07 to 1.18 |
| Quintile 5 (highest) | 0.68 | 0.34 to 1.34 | 0.96 | 0.48 to 1.93 | 0.62 | 0.36 to 1.07 | 0.32* | 0.08 to 0.92 |
| N | 3072 | | 3072 | | 3072 | | 1322 | |

The bold font is to emphasize the key independent variable in each table, which is the binary indicator for military camp population.
*P≤0.05, **p≤0.01; analysis for obtaining desired method is limited to women currently using contraception.

**Table 4** Weighted logistic regression results for differences in family planning knowledge among military, Kinshasa 2016

| | Heard of implants | | Heard of injectables | | Heard of condoms | | Heard of withdrawal | | Heard of rhythm | |
|---|---|---|---|---|---|---|---|---|---|---|
| | OR | 95% CI | OR | 95% CI | OR | 95% CI | OR | 95% CI | OR | 95% CI |
| Military | **0.23**\*\* | **0.11 to 0.48** | **0.19**\*\* | **0.08 to 0.44** | **0.23**\*\* | **0.12 to 0.47** | **0.05**\*\* | **0.02 to 0.17** | **0.12**\*\* | **0.03 to 0.44** |
| Number of births | 1.06 | 0.95 to 1.19 | 1.27\*\* | 1.08 to 1.50 | 0.83\* | 0.72 to 0.97 | 1.09 | 0.91 to 1.32 | 1.11 | 0.93 to 1.34 |
| Age | 1.49\*\* | 1.31 to 1.70 | 1.31\*\* | 1.20 to 1.43 | 1.47\*\* | 1.26 to 1.72 | 1.27\*\* | 1.12 to 1.45 | 1.57\*\* | 1.39 to 1.77 |
| Age squared | 0.99\*\* | 0.99 to 0.99 | 0.99\*\* | 0.99 to 0.99 | 0.99\*\* | 0.99 to 0.99 | 0.99\*\* | 0.99 to 0.99 | 0.99\*\* | 0.99 to 0.99 |
| Level of education | | | | | | | | | | |
| No education (ref.) | – | – | – | – | – | – | – | – | – | – |
| Primary | 3.01\*\* | 1.40 to 6.45 | 1.12 | 0.64 to 1.95 | 0.36 | 0.07 to 1.90 | 3.14\*\* | 1.58 to 6.24 | 2.40\*\* | 1.30 to 4.41 |
| Middle secondary school | 5.51\*\* | 2.32 to 13.10 | 3.15\*\* | 1.60 to 6.22 | 0.97 | 0.13 to 7.18 | 7.85\*\* | 3.11 to 19.85 | 4.29\*\* | 1.97 to 9.31 |
| Advanced secondary or higher | 10.85\*\* | 4.11 to 28.62 | 6.23\*\* | 2.34 to 16.56 | 6.57 | 0.55 to 78.84 | 27.43\*\* | 5.77 to 130.44 | 12.80\*\* | 3.74 to 43.81 |
| Marital status | | | | | | | | | | |
| Married/partnered (ref.) | – | – | – | – | – | – | – | – | – | – |
| Separated/divorced/widowed | 1.18 | 0.56 to 2.48 | 0.96 | 0.39 to 2.36 | 0.97 | 0.24 to 3.87 | 1.21 | 0.63 to 2.34 | 1.44 | 0.57 to 3.64 |
| Never married | 0.65\* | 0.46 to 0.94 | 0.89 | 0.56 to 1.41 | 1.27 | 0.58 to 2.75 | 0.95 | 0.68 to 1.33 | 0.63 | 0.37 to 1.08 |
| Household wealth | | | | | | | | | | |
| Quintile 1 (lowest, ref.) | – | – | – | – | – | – | – | – | – | – |
| Quintile 2 | 0.95 | 0.48 to 1.88 | 0.83 | 0.45 to 1.54 | 0.75 | 0.27 to 2.07 | 0.38\*\* | 0.18 to 0.79 | 0.41\* | 0.18 to 0.92 |
| Quintile 3 | 0.79 | 0.39 to 1.62 | 0.86 | 0.42 to 1.75 | 0.99 | 0.38 to 2.60 | 0.45\* | 0.20 to 0.98 | 0.48 | 0.21 to 1.14 |
| Quintile 4 | 0.68 | 0.33 to 1.43 | 0.86 | 0.44 to 1.70 | 0.65 | 0.27 to 1.53 | 0.77 | 0.31 to 1.93 | 0.60 | 0.26 to 1.38 |
| Quintile 5 (highest) | 0.72 | 0.30 to 1.71 | 0.64 | 0.28 to 1.48 | 1.06 | 0.37 to 3.01 | 0.60 | 0.25 to 1.47 | 0.41\* | 0.19 to 0.93 |
| N | 3078 | | 3077 | | 3080 | | 3079 | | 3077 | |

The bold font is to emphasize the key independent variable in each table, which is the binary indicator for military camp population.
*P≤0.05, **p≤0.01.

**Table 5** Weighted logistic regression results for differences in FP exposure among military, Kinshasa 2016

| | Visited health facility | | Heard about FP–radio | | Read about FP | | FP billboard | |
|---|---|---|---|---|---|---|---|---|
| | OR | 95% CI | OR | 95% CI | OR | 95% CI | OR | 95% CI |
| Military | **0.28**\** | **0.14 to 0.55** | **2.81**\** | **1.34 to 5.92** | **7.72**\** | **3.13 to 19.04** | **9.71**\** | **3.63 to 25.20** |
| Number of births | 1.07 | 1.00 to 1.15 | 0.96 | 0.90 to 1.03 | 0.86* | 0.77 to 0.97 | 0.97 | 0.90 to 1.05 |
| Age | 1.11** | 1.04 to 1.17 | 1.11* | 1.03 to 1.20 | 1.12 | 1.00 to 1.25 | 1.11** | 1.05 to 1.18 |
| Age squared | 0.99** | 0.99 to 0.99 | 1.00 | 0.99 to 1.00 | 1.00 | 0.99 to 1.00 | 0.99** | 0.99 to 1.00 |
| Level of education | | | | | | | | |
| No education (ref.) | – | – | – | – | – | – | – | – |
| Primary | 1.21 | 0.60 to 2.42 | 3.30** | 1.52 to 7.18 | 3.17 | 0.96 to 10.50 | 1.22 | 0.60 to 2.56 |
| Middle secondary school | 1.36 | 0.67 to 2.76 | 6.00** | 2.33 to 15.48 | 6.05** | 1.80 to 20.34 | 1.34 | 0.75 to 2.48 |
| Advanced secondary or higher | 1.91 | 0.85 to 4.28 | 13.93** | 4.88 to 39.74 | 15.99** | 4.26 to 60.00 | 2.18* | 1.12 to 4.30 |
| Marital status | | | | | | | | |
| Married/partnered (ref.) | – | – | – | – | – | – | – | – |
| Separated/divorced/widowed | 1.05 | 0.62 to 1.76 | 0.89 | 0.57 to 1.38 | 1.00 | 0.55 to 1.80 | 0.94 | 0.56 to 1.66 |
| Never married | 0.53** | 0.42 to 0.68 | 0.84 | 0.59 to 1.18 | 0.82 | 0.55 to 1.24 | 1.03 | 0.74 to 1.42 |
| Household wealth | | | | | | | | |
| Quintile 1 (lowest, ref.) | – | | – | – | – | – | – | – |
| Quintile 2 | 0.52** | 0.33 to 0.83 | 1.05 | 0.55 to 2.00 | 0.81 | 0.36 to 1.85 | 1.03 | 0.62 to 1.69 |
| Quintile 3 | 0.55* | 0.35 to 0.88 | 0.86 | 0.43 to 1.72 | 0.58 | 0.25 to 1.36 | 1.08 | 0.63 to 1.94 |
| Quintile 4 | 0.48** | 0.31 to 0.75 | 0.84 | 0.43 to 1.66 | 0.47 | 0.18 to 1.20 | 0.85 | 0.53 to 1.49 |
| Quintile 5 (highest) | 0.59* | 0.37 to 0.95 | 0.71 | 0.30 to 1.69 | 0.73 | 0.24 to 2.20 | 0.91 | 0.50 to 1.77 |
| N | 3079 | | 3072 | | 3042 | | 3072 | |

The bold font is to emphasize the key independent variable in each table, which is the binary indicator for military camp population.
*P≤0.05, **p≤0.01.
FP, family planning.

The lack of access does not, however, explain the lower use or traditional methods and the greater exposure to family planning messaging among the military camp population. This may be explained by the timing of family planning programmes. According to our authors, new family planning programmes were advertised in military camps recently prior to our data collection, but this was not yet matched by supply of family planning methods in military camp clinics.

This study also serves as a baseline for a programmatic initiative that started shortly after the survey in Kinshasa, which consisted of training clinical personnel at the military health centre in all contraceptive methods, including Implanon NXT (highly popular in Kinshasa); training and supplying community-health workers with a non-medical profile to distribute pills, condoms and CycleBeads at the community level and placing billboards promoting family planning near military camps that showed the father in uniform. A follow-up survey is scheduled for 2018 to assess change in family planning access and use.

The strengths of this study are the rare opportunity to learn more about a neglected, but important, population for family planning; and the representative data for residents of military camps and general population of Kinshasa. However, there remain some limitations. Several measures of interest are not included in our survey instrument. Many military members and their families are likely in-migrants to Kinshasa from elsewhere in DRC; research in this setting has shown that migrants differ from Kinshasa-born residents in many characteristics related to family planning.[29] This study, unfortunately, did not measure migration status, birthplace or duration resided in Kinshasa. Similarly, measures of access to a facility that provides family planning, such as distance to the nearest family planning clinic or pharmacy that has modern methods in stock, were not measured for the military camp population, but may differ from the general population—as suggested by our results.

**Contributors** PA, PA and JB initially conceived the manuscript. PA and PA conducted the statistical analysis and wrote the first draft of the paper. HNE, GKK, JH, PK and JB reviewed the paper before submission and provided comments and edits.

**Funding** PMA2020 was supported by the Bill & Melinda Gates Foundation, under grant #OPP1079004. Data collection for military camps in Kinshasa was also supported by the Bill & Melinda Gates Foundation, under grant # OPP1128892.

**Competing interests** None declared.

**Patient consent** Obtained.

**Ethics approval** This study has received approval to collect data from Institutional Review Boards at Johns Hopkins University, Tulane University and the University of Kinshasa.

**Provenance and peer review** Not commissioned; externally peer reviewed.

**Data sharing statement** All data collected under PMA2020 are made publicly available. More details on data access for PMA2020 are available at: http://www.pma2020.org/data-use.

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
