## [Reviewer comments · BMJ Open]

ARTICLE DETAILS

TITLE (PROVISIONAL)	Differences in Family Planning Outcomes between Military and General Populations in Kinshasa, Democratic Republic of Congo: A Cross-Sectional Analysis
AUTHORS	Akilimali, Pierre; Anglewicz, Philip; Engale, Henri; Kurhenga, Gilbert; Hernandez, Julie; Kayembe, Patrick; Bertrand, Jane

VERSION 1 – REVIEW

REVIEWER	Onaedo Ilozumba Vrije Universiteit, Amsterdam, Netherlands
REVIEW RETURNED	03-Apr-2018

GENERAL COMMENTS	The authors present an overall clear paper on a neglected target group (women living in military camps). However, there are some ethical considerations and structural issues which would improve the readability of the manuscript 1) The authors utilize references from the US context which raises a lot of questions considering that this research study is conducted in DRC. Are the authors sure that more relevant contextual information does not exist? If that is the case then it is also very important to present the information now in the fifth paragraph earlier in the introduction.2) The authors define their age group as 15-49 but no information is provided on the ethical approval process for minors/adolescents3) Age is only presented as a mean, it would be interesting to see the breakdown in age groups. It would also be interesting to explicitly discuss differences or similarities across age in the results4) The authors should clearly define "military women" in their methodology. They should always use consistent language as it now moves between women in military camps, military populations and women in the military population5) What are the traditional contraceptive methods which are mentioned multiple times in the paper6) It would be helpful to see the survey utilized as well the constructs that contribute to the wealth index. These can be supplementary files/appendix documents.7) Report p-values/ORS in the abstract
---

	8) In general the grammar and writing are clear but there are some odd phrases and typos which some editing would easily address. Examples include "strong and highly statistically significant" , "selected women were administered a survey", etc. 9)There's a mismatch between the title which discuss family planning outcomes and the abstract objective. Please ensure the objective is consistent throughout the paper.
--	--

REVIEWER	Clare Barnett ZEG Berlin, Germany
REVIEW RETURNED	07-Jun-2018

GENERAL COMMENTS	This is a good paper on an area that is clearly under-researched. A few comments  1. I would consider re-ordering the introduction starting with a discussion on the limited published research currently available military access to contraception in developing countries and/or African countries and move onto discussion on why there may be differences between military and non-military populations based on research that is available from US military populations 2. There are many linking phrases throughout e.g. "At the same time" Many of these could be removed to allow for a tighter reading of the text 3. Page 6, Line 22 - why is the sample of female military members so much higher than the percentage of women in the military (16% vs 3%). Is this an artefact of sampling or is there something else going on? 4. Page 6, Line 36 - there seems to be something missing. The phrase "whether the woman experienced the death of one of her children do not want another child" does not make sense 5. Page 8, there is inconsistency in reporting of decimal places (sometimes 2 decimals, sometimes 3 decimals)
--

VERSION 1 – AUTHOR RESPONSE

Reviewers' Comments to Author:

Reviewer: 1

The authors present an overall clear paper on a neglected target group (women living in military camps). However, there are some ethical considerations and structural issues which would improve the readability of the manuscript.

1) The authors utilize references from the US context which raises a lot of questions considering that this research study is conducted in DRC. Are the authors sure that more relevant contextual information does not exist? If that is the case then it is also very important to present the information now in the fifth paragraph earlier in the introduction.

To verify that we were not missing relevant literature on this topic, we again searched research databases like Google scholar and Pubmed (using search terms like "military family planning," "military contraceptive use," "military family planning Africa," "military contraceptive use Africa"), and we did not find any research similar to our study. The vast majority of research on this topic comes from the United States, which illustrated the need for more studies on family planning among military in an international context.

We appreciate the suggestion to move up the fifth paragraph of the paper. We now present this information in the first paragraph.

2) The authors define their age group as 15-49 but no information is provided on the ethical approval process for minors/adolescents.

For this research we follow international procedures that permit each country to determine the age of majority. In DRC, individuals aged 15-17 are considered adults, and it is therefore not necessary to obtain parental approval for them to be interviewed.

To clarify this issue, we have added the following information to the paragraph on ethical approval for the study “All participating women provided written and informed consent to take part in this study. In this setting, individuals aged 15-17 are considered adults, so parental consent was not necessary to interview women of these ages” (pg. 6).

3) Age is only presented as a mean, it would be interesting to see the breakdown in age groups. It would also be interesting to explicitly discuss differences or similarities across age in the results.

We have added the percentage at each five year age group for the military and non-military populations in Table 1, replacing the mean age.

We also tested for significant differences in the proportion at each five year age group by these populations, and none of these tests were statistically significant (at $p < 0.05$ or lower). We also re-ran our regression models using five year age groups instead of the mean and quadratic terms, and the results are not substantively different.

Because (1) the only statistically significant difference between the population is for the mean age and not for the five year age groups, (2) and the regression results are the same for five year age groups and mean age, we do not include five year age groups in the analysis shown in this paper, and only show the five year age groups in Table 1.

4) The authors should clearly define "military women" in their methodology. They should always use consistent language as it now moves between women in military camps, military populations and women in the military population.

To be clear and consistent in describing our population, we use the phrase “women in military camps” throughout this paper.

5) What are the traditional contraceptive methods which are mentioned multiple times in the paper?

The traditional methods included in the survey instrument are “rhythm method”, “withdrawal”, and “other traditional method” (folkloric methods like amulets, herbs, etc...). We now list these traditional methods on pg. 6.

6) It would be helpful to see the survey utilized as well the constructs that contribute to the wealth index. These can be supplementary files/appendix documents.

We have included the survey instruments as appendix documents.

7) Report p-values/ORS in the abstract.

We have included odds ratios and 95% confidence intervals in the abstract.

8) In general the grammar and writing are clear but there are some odd phrases and typos which some editing would easily address. Examples include "strong and highly statistically significant" , "selected women were administered a survey", etc.

We appreciate this suggestion, and have reviewed the paper to improve the writing, including edits to the phrases above.

9) There's a mismatch between the title which discuss family planning outcomes and the abstract objective. Please ensure the objective is consistent throughout the paper.

We have rephrased the abstract objective to more clearly match the title of the paper, and have updated other sections of the paper as well.

Reviewer: 2

This is a good paper on an area that is clearly under-researched.

A few comments

1. I would consider re-ordering the introduction starting with a discussion on the limited published research currently available military access to contraception in developing countries and/or African countries and move onto discussion on why there may be differences between military and non-military populations based on research that is available from US military populations

We appreciate these suggestions and have moved the description of the limited research on this topic to the first paragraph of the paper. We prefer to retain the discussion on why there may be differences between military and non-military populations, since this helps to justify and frame this study.

2. There are many linking phrases throughout e.g. "At the same time" Many of these could be removed to allow for a tighter reading of the text.

We have reviewed the paper to remove many of the linking phrases described above, and to generally improve the writing.

3. Page 6, Line 22 - why is the sample of female military members so much higher than the percentage of women in the military (16% vs 3%). Is this an artefact of sampling or is there something else going on?

In this case, the numerators and denominators are different in the percentages being compared. In the first, the percentage of all military members who are female (3%), the numerator is the number of women, and the denominator is the number of all members of the military. In the second (16%), the numerator is the number of women in the military and the denominator is the number of women who reside in military camps. So in short, the percentage of female military camp residents who are in the military is larger than the percentage of military members who are female.

4. Page 6, Line 36 - there seems to be something missing. The phrase "whether the woman experienced the death of one of her children do not want another child" does not make sense

We have corrected this error, and the sentence now states “We focus on several categories of family planning outcomes, starting with four fertility-related outcomes: the number of lifetime births, whether the woman experienced the death of one of her children, whether the women does not want another child, and whether the last birth was unintended.”

5. Page 8, there is inconsistency in reporting of decimals places (sometimes 2 decimals, sometimes 3 decimals)

We have removed the third digit past the decimal, and results now show two decimals throughout.

VERSION 2 – REVIEW

REVIEWER	Clare Barnett ZEG Berlin, Berlin Germany
REVIEW RETURNED	13-Jul-2018

GENERAL COMMENTS	Table 1 has only percentages to the single decimal place - for consistency throughout the paper this should be changed to 2 decimal places.
---

VERSION 2 – AUTHOR RESPONSE

Reviewer's Comments to Author:

Reviewer: 2

Reviewer Name: Clare Barnett

Institution and Country: ZEG Berlin, Berlin Germany

Please state any competing interests or state 'None declared': None declared

Table 1 has only percentages to the single decimal place - for consistency throughout the paper this should be changed to 2 decimal places.

We have added another digit to all numbers in Table 1, so that there are two decimal points total.